# How air pollution affects corporate total factor productivity?

**Jialiang Yang**[1], **Wen Yin**[2]*

1 Jingjiang College, Jiangsu University, Zhenjiang, 212000, Jiangsu, China, 2 Jiangsu College of Tourism, Yangzhou, 225000, Jiangsu, China

* yinw@jstc.edu.cn

**Data Availability Statement:** All relevant data are within the manuscript and its Supporting Information files.

**Funding:** This research was funded by [Top-notch Academic Programs Project of Jiangsu Higher Education Institutions (Provincial First-class

## Abstract

To explore the relationship between air pollution and total factor productivity and new pathways, This paper examines the impact of air pollution on total factor productivity of A-share listed companies in Shanghai and Shenzhen between 2015 and 2019. It investigates this relationship by considering two pathways: investor sentiment and government attention. The findings indicate that air pollution suppresses total factor productivity of firms. However, air pollution stimulates investor sentiment, which in turn increases R&D investment and total factor productivity, reducing to some extent the dampening effect of air pollution on total factor productivity. There exists a notable positive correlation between air quality and government attention, acting as a mediating variable. This implies that air pollution has the potential to capture the attention of governmental entities, leading to the implementation of appropriate measures aimed at managing and mitigating the occurrence of air pollution caused by industrial enterprises.And the relevant governments should formulate a series of policies to meet the different needs of different enterprises. These two approaches have varying impacts depending on the type of enterprises, thus governments should develop laws to cater to the various demands of different types of enterprises.

## Introduction

With the intention of transforming the global governance framework for sustainable development, the United Nations Development Summit put out 17 global sustainable development goals in the 2030 Agenda for Sustainable Development. Achieving sustainable growth will inevitably demand raising total factor productivity. China is the greatest manufacturing nation in the world, and as such, industrial pollution has led to extreme air pollution, which is a major obstacle to the countr"s sustainable social and economic growth. Thus, it is crucial to investigate if Chin"s total factor productivity has been hampered by air pollution. Some academics have been studying total factor productivity recently. The influencing factors of total factor productivity (TFP) vary across different sectors and regions, reflecting a complex interplay of economic, environmental, and policy-related variables. For urban agglomerations, green total factor productivity (GTFP) is influenced by technological progress, industrial structure, financial service level, and the business environment, highlighting the importance of

Specialty Construction of Energy Economy in Jingjiang College of Jiangsu University), Award number: 020106T], It provides financial support for this paper. The funders had no role in study design, data collection and analysis, decision to publish, or preparation of the manuscript

**Competing interests:** The authors have declared that no competing interests exist.

regional synergistic development strategies [1]. In the context of the Yellow River basin, fiscal decentralization, industrial structure, financial development, urbanization level, and research and development investment are key drivers of urban ecological TFP [2]. For Chin"s industries, innovation investment, urbanization level, FDI, and environmental governance are among the determinants of GTFP, with spatial and temporal heterogeneity in their effects [3].

The impact of air pollution on enterprises is multifaceted, influencing various aspects of their operations, productivity, and strategic decisions. Research indicates that air pollution significantly decreases enterprise-level energy efficiency, affecting productivity, increasing total energy consumption, and lowering exports, with varying impacts based on enterprise characteristics such as ownership, age, and location [4]. In response to air quality challenges, some enterprises are motivated to upgrade to industrial intelligence, utilizing intelligent industrial robots to mitigate labor shortages and inefficiencies caused by pollution [5]. Air pollution also negatively affects entrepreneurial activities, reducing the number of newly registered firms by about 36%, influenced by factors like brain drain, social capital, startup costs, and financial constraints [6]. It inhibits firm productivity directly and through spatial spillover effects, with the deterioration of air quality impacting research and development, human capital, and government subsidies [7]. The siting of enterprises is adversely affected by air pollution, discouraging the establishment of businesses in polluted areas due to reduced local labor endowment and market scale [8]. Improvements in air quality have been found to significantly increase enterprise productivity, especially after the implementation of more stringent air pollution control measures, highlighting the importance of environmental regulation for economic development [9].

The research of these scholars has important reference value for in-depth investigation of the impact of air pollution on the total factor productivity of enterprises. However, there is still limited literature on the impact of environmental pollution on the total factor productivity of micro enterprises, which cannot provide comprehensive and sufficient decision-making support for governments and enterprises. Based on this, the main work and marginal contributions of this paper include: (1) proposing the effect of air pollution on total factor productivity at the enterprise level, discovering two new impact paths of investor sentiment and government attention, expanding the theoretical framework of existing research, and supplementing and deepening existing research. (2) This paper matches the data of listed companies with the air pollution data of urban administrative regions, improving the accuracy of conclusions through fine matching, and striving to obtain more general conclusions through a large sample size.

## Literature review

Nowadays, labor and capital—two important components of the production factor combination—are the primary subjects of study on the effects of air pollution on firm productivity. From the perspective of labor quality, Air pollution significantly harms both an individua"s physical and mental health. Research has consistently shown that exposure to air pollutants, such as particulate matter (PM10 and PM2.5), sulfur oxides (sOx), carbon monoxide (CO), and nitrogen oxides (nOx), is linked to a range of physical health issues [10]. The mental health impacts of air pollution are increasingly recognized, with evidence suggesting that poor air quality is associated with poor mental health outcomes, including specific mental disorders [11]. These studies collectively emphasize the widespread harm of air pollution to physical and mental health. From the perspective of human capital output, Air pollution negatively affects migration decisions, with a notable decrease in the probability of migrants moving into cities as PM2.5 levels increase, suggesting a preference for cities with better environmental quality to

minimize exposure to air pollution [12]. Skilled workers are more likely to emigrate from polluted areas, leading to a spatial redistribution of labor that affects the supply of skilled versus unskilled workers across cities [13]. This is particularly true for younger and educated workers, exacerbating the drain of a qualified labor force from polluted areas [14]. From the perspective of production cost, Studies have shown that air pollution significantly decreases enterprise-level energy efficiency by reducing enterprise productivity, increasing total energy consumption, and lowering exports[15]. The negative impact of air pollution extends to inhibiting firm productivity, with the spatial spillover effects of pollution from surrounding cities also dampening firm productivity [7]. Moreover, air pollution has been found to negatively affect labor productivity in the industrial sector, with reductions in PM10 levels leading to increases in marginal labor productivity [16]. However, some scholars believe that air pollution may also encourage enterprises to engage in green innovation [17]. By increasing the total number of green patent applications, particularly in the form of patents for green utility models, air pollution significantly promotes enterprises to engage in green innovation [18]. Government regulation plays a crucial role in this dynamic, and research results support the Porter hypothesis, which suggests that environmental regulation significantly improves the green innovation level of enterprises, especially non-state-owned enterprises [19]. Air pollution has been proven to have a positive impact on corporate green management behavior, and media attention amplifies this impact by increasing public scrutiny of corporate pollution [18].

In summary, air pollution may suppress the improvement of enterprise productivity through human resources, production costs, and other means, but it may also force enterprises to engage in green innovation and have a positive effect on the industry. Based on this, it is necessary to examine the impact of air pollution on the total factor productivity of enterprises. Therefore, hypothesis 1 of this paper is proposed: Air quality is negatively correlated with firm" total factor productivity.

Further propose two possible new ways in which air pollution affects the total factor productivity of enterprises.

Hypothesis 2: Air pollution affects total factor productivity of enterprises by increasing investor sentiment.

In terms of air quality and investor sentiment, when investors are in an environment with severe air pollution, they will generate negative emotions, which is detrimental to stock liquidity [20]. Air quality can have a negative impact on investor behavior and may lead to a decrease in trading willingness [21]. Air quality can also affect the investment and judgment of securities market participants through policy, information, and public opinion channels [22]. This is further supported by findings in the realm of corporate finance, where air pollution influences corporate cash holdings and financial leverage, indicating that financial professionals in polluted areas might adjust their strategies based on the environmental conditions, potentially due to the cognitive effects of air pollution [23]. The negative impact of air pollution on enterprise productivity, through mechanisms such as reduced innovation capacity and human capital, further underscores the potential for air quality to impair professional abilities in analytical and judgment-intensive roles [24]. Business managers will have negative expectations for future environmental legislation when air pollution levels are high, which will result in lower investment spending [25]. The emotions of corporate executives can also have an impact on a compan"s research and development investment [26].

Hypothesis 3: Air pollution affects the total factor productivity of enterprises by attracting government attention.

The serious air quality problems in China have aroused deep concern from the government [27]. In China, with the increasing environmental awareness of the public [28], the government is facing enormous pressure to maintain environmental legislation. Most scholars believe

that the financial subsidies brought about by government attention are beneficial, that is, government subsidies can improve the total factor productivity of enterprises [29], and the significant increase in total factor productivity of high-tech enterprises by government subsidies has also been verified [30]. However, Some scholars hold different opinions, believing that the impact of government subsidies on different regions is different, and even subsidies in some regions will not have an impact on the improvement of total factor productivity [31]. Therefore, it is necessary to verify whether this path is significant.

## Methodology

In order to verify hypothesis 1, the following model is established to test the relationship between enterprise air quality and total factor productivity.

$$tfp_{i\tau t} = \beta_0 + \beta_1 aqi_{it} + \sum \beta_j X_{jit} + \sum year + \sum ind + \varepsilon_{it} \tag{1}$$

where $tfp_{i\tau t}$ denotes the total factor productivity of firm $\tau$ in year $t$ in region $i$, $aqi_{it}$ denotes the level of air pollution in year $t$ in region $i$, X denotes firm-level and macro-level control variables, $\sum year$ is a time fixed effect, $\sum ind$ is an industry fixed effect, and $\varepsilon_{it}$ denotes the error term.

This study investigates the potential impact pathways between air quality and total factor productivity of firms, building upon earlier research in this area. The analysis use a mediating effects model to explore these routes. Two sets of mediating effect models have been created in order to assess research hypothesis 2 and hypothesis 3:

$$sent_{i\tau t} = \alpha_0 + \alpha_1 aqi_{it} + \sum \alpha_j X_{jit} + \sum year + \sum ind + \varepsilon_{it} \tag{2}$$

$$crd_{i\tau t} = \theta_0 + \theta_1 aqi_{it} + \theta_2 sent_{i\tau t} + \sum \theta_j X_{jit} + \sum year + \sum ind + \varepsilon_{it} \tag{3}$$

$$tfp_{i\tau t} = \gamma_0 + \gamma_1 aqi_{it} + \gamma_2 sent_{i\tau t} + \gamma_3 crd_{i\tau t} + \sum \gamma_j X_{jit} + \sum year + \sum ind + \varepsilon_{it} \tag{4}$$

$$gs_{i\tau t} = \alpha_0 + \alpha_1 aqi_{it} + \sum \alpha_j X_{jit} + \sum year + \sum ind + \varepsilon_{it} \tag{5}$$

$$crd_{i\tau t} = \theta_0 + \theta_1 aqi_{it} + \theta_2 gs_{i\tau t} + \sum \theta_j X_{jit} + \sum year + \sum ind + \varepsilon_{it} \tag{6}$$

$$tfp_{i\tau t} = \gamma_0 + \gamma_1 aqi_{it} + \gamma_2 gs_{i\tau t} + \gamma_3 crd_{i\tau t} + \sum \gamma_j X_{jit} + \sum year + \sum ind + \varepsilon_{it} \tag{7}$$

where $sent_{i\tau t}$ denotes investor sentiment in year $t$ of firm $\tau$ in region $i$, $gs_{i\tau t}$ denotes government attention in year $t$ of firm $\tau$ in region $i$, $crd_{i\tau t}$ denotes corporate R&D investment in year $t$ of firm $\tau$ in region $i$. The meanings of the other variables are the same as in equation Eq (1) in which $\beta_1$ is significant is the premise of the test for mediating effects. Eqs (2)–(4) are tests of hypothesis 2 and Eqs (5)–(7) test hypothesis 3. Stepwise regressions of the two sets of mediating models in Eqs (2)–(4) and Eqs (5)–(7) are conducted respectively, focusing on whether $\alpha_1$, $\theta_2$, $\gamma_3$ and $\gamma_1$ are significant, and if all are significant and the $\alpha_1 \times \theta_2 \times \gamma_3$ product has the same sign as $\gamma_1$, hypotheses 2 and 3 hold.

## Variables and data

The relevant variables involved in this paper and their symbols and definitions all are described in detail in Table 1.

**Table 1. Representative symbols and definitions for each variable.**

| Variables | Variable symbols | Definition |
|---|---|---|
| Total factor productivity of enterprises | TFP | Total factor productivity by the LP method |
| Investment in R&D by listed companies | CRD | Annual R&D expenses/principal income of listed companies |
| Air Quality | AQI | Average of the daily air quality index for the year in the city where the listed company is located |
| Investor sentiment | Sent | Semi-annual momentum indicator, i.e. cumulative monthly stock returns for In the previous year 7 to 12 months |
| Government Concerns | GS | Total annual government subsidies for listed companies |
| Gearing ratio | ALE | Total liabilities at end of period / Total assets at end of period |
| Size of business | Size | Natural logarithm of total assets at the end of the period |
| Cash holdings | Cash | Amount of cash and short-term investments at the end of the period/total assets |
| Free cash flow | FCF | Cash flow from operating activities / balance sheet total at the end of the financial year |
| Technical complexity | TCD | Screening and comparing different regional innovation systems to reflect regional technological competitive advantage |
| City year-end population | CYP | Total population at the end of the year in the cities where listed companies are located. |
| GDP growth rate | GDP | The GDP growth of the listed company's local city |
| Key pollution monitoring dummy variables | Control | 1 for key pollution regulated enterprises, 0 for the rest |
| Dummy variables for state-owned enterprises | State | 1 for state-owned enterprises, 0 for the rest |
| Dummy variables for polluters | Pollute | 1 for polluters, 0 for the rest |

## Explanatory variable-total factor productivity (TFP) of firms

All companies measuring TFP are all A-share listed companies in China from 2015–2019. In terms of measuring total factor productivity, the LP method is an improved version of the OP method, which not only solves the problem of invalid samples due to zero investment data, but also effectively mitigates endogenous problems such as directional causality, sample self-selection, and linkage bias, while retaining the advantages of the OP method [32]. This paper therefore selects the LP method firm total factor productivity.

## Core explanatory variable-Air Quality (AQI)

Drawing on the air pollution indicators constructed by Xue Shuang et al. (2017) in the area of investment arbitrage [33], this document measures air pollution using the average daily air quality index of the city in which the company is located for the year.
    Mediating variables

## Mediating variables

Investor sentiment (Sent): This paper draws on Polk and Sapienza's design of momentum indicators [34], the momentum effect is transient and usually lasts from 3 to 12 months, while momentum indicators in China's stock market are significantly present in the semi-annual

period, while they reverse after the semi-annual period or longer, so this paper uses semi-annual momentum indicators as a proxy variable for investor sentiment for the follow-up study [35], Sent, a cumulative monthly yield that considers cash dividends with a six-month lag, is used to measure investor sentiment.

Government attention (GS): For micro-level firms, drawing on the calculation of regional government R&D attention, this document uses total government subsidies to companies to represent the government's focus [36].

Corporate R&D investment (CRD) This paper refers to Hansen et al [37] and uses company R&D expenditure/main business revenue to calculate.

### Control variables

Based on studies by related scholars [38], this paper, we choose as control variables the mesh (ALE), firm size (Size), cash holdings (Cash), free cash flow (FCF), technical complexity (TCD), city year-end population (CYP) and GDP growth (GDP).

### Dummy variables

This study used heterogeneity analysis by utilizing dummy variables. Key pollution monitoring enterprise is 1, the rest is 0. State-owned enterprise is 1, the rest is 0. Polluting enterprise is 1, the rest is 0.

### Data

This study focuses on A-share listed companies in Shanghai and Shenzhen as the research sample. The air quality index (AQI) data is obtained from China's Ministry of Ecology and Environment. The sample data for enterprises during the period of 2015–2019 is selected based on data availability. The data in this document are mainly derived from the Guotaian CSMAR database, macroeconomic data from the China Ur-ban Statistical Yearbook and patent data from the State Intellectual Property Office on the technical complexity of cities, descriptive statistical analysis is shown in Table 2.

## Results and discussion

In this paper, total factor productivity (TFP) has a minimum value of 1.033 and a maximum value of 12.301, with a standard deviation of 1.296, which can be seen in the fact that firms are not the same and have a more pronounced difference in their TFP. The average value of the AQI is 81.625.

### Baseline regression analysis

Table 3 shows the baseline regression results of the impact of air quality on total factor productivity of enterprises. We can see that controlling for the variables of gearing, total year-end population, economic growth rate, Size of company, cash stock, free cash flow and technical complexity, the regression coefficients of the air quality index (AQI) are at this point significantly negative, with the following economic implications: in model (1), A regression coefficient of -0.038, for example, implies a one-unit increase in the air quality index (AQI). At the beginning, the decline rate of total factor production of listed companies reached 3.8%, hypothesis 1 is therefore tested and confirmed.

Regarding the other variables, firm size (Size) is significantly and positively related to firm total factor productivity (TFP), so it is known that firms with good development size have higher firm total factor productivity.

**Table 2. Descriptive statistics.**

| Variables | Mean | sd | Min | Max |
|---|---|---|---|---|
| TFP | 7.917 | 1.296 | 1.033 | 12.301 |
| AQI | 81.625 | 19.284 | 0.080 | 247.350 |
| CRD | 2.10E+08 | 8.70E+08 | 0.00E+00 | 2.20E+10 |
| Sent | -0.005 | 0.333 | -1.055 | 13.197 |
| GS | 5.00E+07 | 2.50E+08 | 0.00E+00 | 1.10E+10 |
| TCD | -0.976 | 2.527 | -6.604 | 6.792 |
| ALE | 0.086 | 0.061 | -0.374 | 0.615 |
| CYP | 680.092 | 486.942 | 0 | 3403.641 |
| GDP | 1.20E+04 | 1.00E+04 | 1.53E+02 | 3.80E+04 |
| Size | 9.80E+09 | 4.70E+10 | 8.70E+06 | 1.90E+12 |
| Cash | 44.352 | 27.353 | 0 | 100.01 |
| FCF | 0.345 | 0.367 | 0 | 28.548 |

**Table 3. Basic regression.**

| Variables | (1)TFP | (2)TFP | (3)TFP |
|---|---|---|---|
| | POLS | FE | RE |
| AQI | -0.038*** | -0.060*** | -0.039*** |
| | (-4.09) | (-4.97) | (-4.00) |
| ALE | 0.272*** | -0.096*** | 0.001 |
| | (8.77) | (-3.03) | (0.05) |
| Size | 1.509*** | 1.244*** | 1.390*** |
| | (114.49) | (55.92) | (85.88) |
| Cash | -0.037 | -0.196*** | -0.165*** |
| | (-0.68) | (-4.70) | (-4.16) |
| FCF | 1.173*** | 0.444*** | 0.506*** |
| | (17.81) | (11.44) | (13.29) |
| TCD | 0.002 | -0.003*** | -0.003*** |
| | (0.80) | (-2.95) | (-2.90) |
| CYP | 0.006*** | -0.005 | 0.002 |
| | (3.28) | (-0.93) | (0.86) |
| GDP | 0.267** | 0.098 | 0.103 |
| | (2.27) | (1.52) | (1.63) |
| _cons | -6.573*** | -3.518*** | -5.063*** |
| | (-48.81) | (-15.67) | (-32.07) |
| Time effect | Yes | Yes | Yes |
| Industry effects | Yes | Yes | Yes |
| Regional effects | Yes | Yes | Yes |
| N | 10569 | 10569 | 10569 |
| $r^2$ | 0.655 | 0.332 | 0.330 |

Note

***, ** and * denote significant at the 1%, 5% and 10% levels respectively, with standard errors in brackets. The following table is identical.

## Robustness tests

**Endogenous processing.** In order to reduce the influence of endogeneity, a variable closely related to AQI but not directly affecting the total factor productivity of enterprises will be selected as a tool in this paper.

It is known that the higher the wind, the easier it is to disperse the airborne particles, thus reducing the level of air pollution; we can then know that the higher the air flow coefficient, the more fluid the air is, and the faster it can be blown away from the air. Drawing on the study of Ma et al [39], Air flow coefficient (CUR) is a tool for this article. The air flow coefficient was calculated by multiplying the ten-meter wind speed by the height of the boundary layer, returning the results shown in Table 4.

In column (1), the regression coefficient of the first stage air flow coefficient (CUR) is significantly negative, while the F-statistic equals 252.073, thus there is no question of a weak instrumental variable, So the air quality is negative correlation relationship between the air flow coefficient. The results of the second stage verify that after the variables are added, the regression coefficient in Column (2) is significantly positive, which also confirms that air pollution can indeed reduce the total factor productivity of enterprises, and the endogeneity problem is also reduced to a certain extent.

**Table 4. Endogeneity test.**

| Variables | (1)AQI | (2)TFP |
|---|---|---|
| | Tool Variable Phase I | Tool Variables Phase II |
| CUR | -0.056*** | |
| | (-15.88) | |
| AQI | | -0.147** |
| | | (-2.47) |
| ALE | 0.021 | 0.275*** |
| | (0.64) | (8.80) |
| Size | 0.099*** | 1.521*** |
| | (7.10) | (104.12) |
| Cash | 0.203*** | -0.030 |
| | (3.52) | (-0.54) |
| FCF | -0.264*** | 1.152*** |
| | (-3.80) | (16.83) |
| TCD | 0.044*** | -0.001 |
| | (18.06) | (-0.24) |
| CYP | 0.081*** | 0.013*** |
| | (50.48) | (2.61) |
| GDP | -2.353*** | 0.047 |
| | (-19.65) | (0.27) |
| _cons | 2.825*** | -6.388*** |
| | (16.97) | (-40.06) |
| Time effect | Yes | Yes |
| Industry effects | Yes | Yes |
| Regional effects | Yes | Yes |
| F-test (Stage 1) | 252.073 | |
| N | 10569 | 10569 |
| $r^2$ | 0.241 | 0.649 |

## Split time period test

China has proposed relevant policies to significantly reduce PM2.5 concentrations in key areas by 2017. To avoid significant changes in air quality due to this policy, the paper divides the sample regression into two time periods, 2015–2016 and 2017–2019, to do a regression of Eq (1). The results are shown in Table 5, where air quality has a significant negative effect on the total factor productivity at all different time periods.

## Analysis of mechanisms

Column (1) of Table 6 shows that air pollution significantly reduces the total factor productivity in the absence of mediating variables, and column (2) shows that the coefficient of the explanatory variable AQI on the mediating variable investor sentiment (Sent) is significantly positive. In column (3), investor sentiment (Sent) is significantly positive on firms' investment in research and development (CRD), indicating that the volatility generated by aroused investor sentiment causes firms to invest more in R&D. Meanwhile, the results in column (4) indicate that corporate R&D investment (CRD) increases corporate total factor productivity (TFP) at the 1% level, which could suggest a mediating effect of investor sentiment and corporate R&D investment between air quality and corporate total factor productivity. All in all, if the production of enterprises pollutes the atmosphere, it will hinder the enhance of the total factor productivity, because the air pollution will arouse the awareness of reducing pollution emissions and take corresponding actions, thus improving the total factor productivity.

**Table 5. Robustness tests by time period.**

| Variables | (1)TFP | (2)TFP |
|---|---|---|
| | Year: 2015–2016 | Year: 2017–2019 |
| AQI | -0.044*** | -0.033*** |
| | (-3.13) | (-2.59) |
| ALE | 0.253*** | 0.283*** |
| | (4.86) | (7.30) |
| Size | 1.512*** | 1.510*** |
| | (65.92) | (93.57) |
| Cash | -0.108 | -0.007 |
| | (-1.20) | (-0.11) |
| FCF | 1.215*** | 1.144*** |
| | (10.80) | (14.04) |
| TCD | 0.002 | -0.003 |
| | (0.38) | (-0.77) |
| CYP | 0.010*** | 0.001 |
| | (3.33) | (0.55) |
| GDP | 0.634** | 0.068 |
| | (2.49) | (0.50) |
| _cons | -6.550*** | -6.537*** |
| | (-28.67) | (-38.98) |
| Time effect | Yes | Yes |
| Industry effects | Yes | Yes |
| Regional effects | Yes | Yes |
| N | 3808 | 6761 |
| $r^2$ | 0.639 | 0.664 |

**Table 6. Sent-CRD mediating effects.**

| Variables | (1) | (2) | (3) | (4) |
|---|---|---|---|---|
| | TFP | Sent | CRD | TFP |
| AQI | -0.038*** | 0.069*** | -0.061 | -0.036*** |
| | (-4.09) | (5.66) | (-1.30) | (-3.91) |
| ALE | 0.272*** | 0.034 | -1.176*** | 0.317*** |
| | (8.77) | (1.08) | (-7.51) | (10.38) |
| Size | 1.509*** | -0.130*** | 4.755*** | 1.327*** |
| | (114.49) | (-5.80) | (71.45) | (84.21) |
| Cash | -0.037 | 0.193*** | 0.730*** | -0.065 |
| | (-0.68) | (4.58) | (2.66) | (-1.23) |
| FCF | 1.173*** | 0.228*** | 3.029*** | 1.052*** |
| | (17.81) | (5.81) | (9.07) | (16.14) |
| TCD | 0.002 | 0.005*** | -0.066*** | 0.005* |
| | (0.80) | (4.08) | (-4.75) | (1.79) |
| CYP | 0.006*** | 0.014*** | 0.069*** | 0.003* |
| | (3.28) | (2.88) | (8.08) | (1.77) |
| GDP | 0.267** | -0.039 | 1.684*** | 0.204* |
| | (2.27) | (-0.59) | (2.83) | (1.77) |
| Sent | | | 0.463*** | 0.004 |
| | | | (3.69) | (0.18) |
| CRD | | | | 0.038*** |
| | | | | (20.22) |
| _cons | -6.573*** | 0.869*** | -45.260*** | -4.840*** |
| | (-48.81) | (3.83) | (-66.58) | (-30.73) |
| Time effect | Yes | Yes | Yes | Yes |
| Industry effects | Yes | Yes | Yes | Yes |
| Regional effects | Yes | Yes | Yes | Yes |
| N | 10569 | 10569 | 10569 | 10569 |
| $r^2$ | 0.655 | 0.024 | 0.391 | 0.668 |

This result verifies hypothesis 2 of the paper. That is, air pollution promotes technological innovation in companies by stimulating investor sentiment, which ultimately improves total factor productivity.

Table 7 shows the results of the baseline regression of the mediating effects of Eqs (5)–(7), with the mediating variables being government concern (GS) and corporate investment in R&D (CRD).The results show that air quality as an explanatory variable has a significantly positive correlation with government attention as a mediating variable, that is, air pollution can attract government attention and relevant measures will be taken to control and control the phenomenon of air pollution generated by enterprises.

At this point, the regression coefficient of government attention (GS) on corporate R&D investment (CRD) in column (3) is significant at the 1% level, indicating that government attention also leads to higher corporate R&D investment, while the regression coefficient of corporate R&D investment (CRD) on corporate total factor productivity (TFP) in column (4) is significantly positive, indicating that there is a mediating effect between the government's focus and corporate R&D investments on air quality and corporate TFP. In general, the increase in air pollution can indeed draw the government's effective attention to implement corresponding subsidy policies, and subsidies can also play an indirect role in increasing R&D investment by listed companies.

**Table 7. GS-CRD mediating effects.**

| Variables | (1) | (2) | (3) | (4) |
|---|---|---|---|---|
| | TFP | GS | CRD | TFP |
| AQI | -0.038*** | 6.207* | -0.032 | -0.036*** |
| | (-4.09) | (1.93) | (-0.75) | (-3.93) |
| ALE | 0.272*** | -9.056 | -0.798*** | 0.316*** |
| | (8.77) | (-1.26) | (-5.56) | (10.34) |
| Size | 1.509*** | 65.731*** | 3.227*** | 1.332*** |
| | (114.49) | (11.79) | (46.49) | (82.22) |
| Cash | -0.037 | -16.380* | 0.665*** | -0.066 |
| | (-0.68) | (-1.73) | (2.66) | (-1.23) |
| FCF | 1.173*** | 33.931*** | 2.238*** | 1.056*** |
| | (17.81) | (3.84) | (7.35) | (16.25) |
| TCD | 0.002 | 1.054*** | -0.056*** | 0.005* |
| | (0.80) | (3.14) | (-4.39) | (1.78) |
| CYP | 0.006*** | 1.821 | 0.045*** | 0.003* |
| | (3.28) | (1.64) | (5.78) | (1.81) |
| GDP | 0.267** | 6.492 | 1.168** | 0.205* |
| | (2.27) | (0.43) | (2.15) | (1.78) |
| GS | | | 0.015*** | -0.000 |
| | | | (45.75) | (-1.35) |
| CRD | | | | 0.039*** |
| | | | | (19.04) |
| _cons | -6.573*** | -608.365*** | -31.289*** | -4.884*** |
| | (-48.81) | (-11.33) | (-45.21) | (-30.37) |
| Time effect | Yes | Yes | Yes | Yes |
| Industry effects | Yes | Yes | Yes | Yes |
| Regional effects | Yes | Yes | Yes | Yes |
| N | 10569 | 10569 | 10569 | 10569 |
| $r^2$ | 0.655 | 0.042 | 0.492 | 0.668 |

It has a positive impact on the increase of total factor productivity of the firm In this regard, Hypothesis 3 was also confirmed.

## Heterogeneity analysis

**Air quality, nature of ownership and total factor productivity of firms.** The nature of the ownership of a listed company (State) is designated as a dummy variable according to the nature of the beneficial owner. The value is 1 for soes and 0 for non-soes.

Table 8 shows the grouped regression results of ownership nature with investor sentiment and corporate R&D investment as mediating variables. Where models (1)-(4) are for the SOE sample and models (5)-(8) are for the non-SOE sample.

The estimated coefficients of the AQI on investor sentiment in models (1)-(4) are not significant, while the estimated coefficients of the AQI on investor sentiment and total factor productivity of firms in models (5)-(8) are all significant. The regression coefficient of investor sentiment on firm innovation R&D investment is significantly positive, as is the regression coefficient of firm R&D investment on total factor productivity. The above results show that investor sentiment mediates total factor productivity in non-state owned enterprises, but investor sentiment in state owned enterprises is not significantly influenced by air quality.

**Table 8. Analysis of heterogeneity in the nature of property rights in relation to investor sentiment.**

| Variables | (1) | (2) | (3) | (4) | (5) | (6) | (7) | (8) |
|---|---|---|---|---|---|---|---|---|
| | Stata = 1 | | | | Stata = 0 | | | |
| | TFP | Sent | CRD | TFP | TFP | Sent | CRD | TFP |
| AQI | -0.118*** | -0.014 | -0.079 | -0.116*** | -0.029*** | 0.073*** | -0.046 | -0.027*** |
| | (-2.73) | (-0.32) | (-0.24) | (-2.73) | (-3.02) | (5.78) | (-0.97) | (-2.88) |
| ALE | 0.002 | 0.165 | -4.503*** | 0.118 | 0.280*** | 0.029 | -0.949*** | 0.317*** |
| | (0.01) | (1.37) | (-4.05) | (0.80) | (8.84) | (0.88) | (-6.12) | (10.16) |
| Size | 1.334*** | -0.138 | 5.930*** | 1.186*** | 1.513*** | -0.130*** | 4.640*** | 1.333*** |
| | (22.76) | (-1.22) | (13.44) | (17.32) | (111.25) | (-5.64) | (69.70) | (81.97) |
| Cash | 0.353 | 0.019 | -0.101 | 0.356 | -0.065 | 0.199*** | 0.671** | -0.091* |
| | (1.39) | (0.10) | (-0.05) | (1.42) | (-1.17) | (4.60) | (2.47) | (-1.68) |
| FCF | 1.187*** | 0.189 | 3.987 | 1.061*** | 1.146*** | 0.229*** | 3.000*** | 1.025*** |
| | (3.56) | (1.16) | (1.57) | (3.19) | (17.11) | (5.68) | (9.11) | (15.46) |
| TCD | 0.007 | 0.018*** | -0.229* | 0.012 | 0.002 | 0.004*** | -0.061*** | 0.005* |
| | (0.39) | (3.20) | (-1.81) | (0.75) | (0.79) | (3.64) | (-4.50) | (1.71) |
| CYP | 0.042*** | 0.049 | 0.313*** | 0.034*** | 0.004** | 0.014*** | 0.057*** | 0.002 |
| | (4.77) | (1.43) | (4.77) | (3.85) | (2.16) | (2.74) | (6.68) | (0.92) |
| GDP | -0.288 | -0.193 | 4.862 | -0.399 | 0.275** | -0.030 | 1.455** | 0.220* |
| | (-0.58) | (-0.89) | (1.31) | (-0.82) | (2.27) | (-0.45) | (2.45) | (1.85) |
| Sent | | | 0.858 | 0.056 | | | 0.448*** | 0.001 |
| | | | (0.81) | (0.40) | | | (3.65) | (0.05) |
| CRD | | | | 0.025*** | | | | 0.039*** |
| | | | | (4.02) | | | | (19.42) |
| Time effect | Yes | Yes | Yes | Yes | Yes | Yes | Yes | Yes |
| Industry effects | Yes | Yes | Yes | Yes | Yes | Yes | Yes | Yes |
| Regional effects | Yes | Yes | Yes | Yes | Yes | Yes | Yes | Yes |
| _cons | -4.992*** | 0.971 | -57.064*** | -3.560*** | -6.593*** | 0.853*** | -44.136*** | -4.880*** |
| | (-8.56) | (0.86) | (-13.00) | (-5.27) | (-47.43) | (3.68) | (-64.87) | (-30.02) |
| N | 473 | 473 | 473 | 473 | 10096 | 10096 | 10096 | 10096 |
| $r^2$ | 0.704 | 0.049 | 0.489 | 0.715 | 0.654 | 0.025 | 0.389 | 0.667 |

Table 9 shows the regression results for the grouping of the nature of property rights with government attention and corporate R&D investment as mediating variables, where models (1)-(4) are for the SOE sample and models (5)-(8) are for the non-SOE sample. The explanatory variable air quality in model (2) is not significant for the mediating variable government concern, indicating that there is no mediating effect in the SOE sample. However, in models (5)-(8), the coefficients of the effect of the explanatory variable air quality on the mediating variable government concern are significantly positive, government concern is significantly positive for firm R&D investment, and R&D investment is also significantly positive for firm total factor productivity.

## Air quality, polluting firms and total factor productivity of firms

The characteristics of the company's own pollution level may influence the mediating effect of the "air quality-total factor productivity" relationship. Therefore, government concern does not play its important role in the relationship between air pollution and firm TFP improvement.

**Table 9. Analysis of government-relevant property rights heterogeneity.**

| Variables | (1) | (2) | (3) | (4) | (5) | (6) | (7) | (8) |
|---|---|---|---|---|---|---|---|---|
| | Stata = 1 | | | | Stata = 0 | | | |
| | TFP | GS | CRD | TFP | TFP | GS | CRD | TFP |
| AQI | -0.118*** | -12.119 | -0.044 | -0.116*** | -0.029*** | 7.367** | -0.032 | -0.027*** |
| | (-2.73) | (-0.65) | (-0.16) | (-2.73) | (-3.02) | (2.27) | (-0.74) | (-2.89) |
| ALE | 0.002 | -26.447 | -2.703*** | 0.110 | 0.280*** | -7.550 | -0.661*** | 0.316*** |
| | (0.01) | (-0.58) | (-2.80) | (0.74) | (8.84) | (-1.05) | (-4.62) | (10.14) |
| Size | 1.334*** | 39.255 | 3.215*** | 1.196*** | 1.513*** | 66.376*** | 3.236*** | 1.337*** |
| | (22.76) | (0.86) | (7.35) | (16.99) | (111.25) | (11.97) | (46.41) | (79.97) |
| Cash | 0.353 | 21.410 | -0.304 | 0.357 | -0.065 | -17.434* | 0.636** | -0.092* |
| | (1.39) | (0.31) | (-0.19) | (1.43) | (-1.17) | (-1.85) | (2.54) | (-1.68) |
| FCF | 1.187*** | 27.527 | 4.442** | 1.068*** | 1.146*** | 34.111*** | 2.184*** | 1.028*** |
| | (3.56) | (0.44) | (2.05) | (3.24) | (17.11) | (3.86) | (7.21) | (15.55) |
| TCD | 0.007 | 4.917* | -0.186* | 0.013 | 0.002 | 0.949*** | -0.052*** | 0.005* |
| | (0.39) | (1.84) | (-1.71) | (0.75) | (0.79) | (2.84) | (-4.13) | (1.70) |
| CYP | 0.042*** | 0.987 | 0.206*** | 0.034*** | 0.004** | 1.755 | 0.038*** | 0.002 |
| | (4.77) | (0.07) | (3.60) | (3.86) | (2.16) | (1.61) | (4.91) | (0.95) |
| GDP | -0.288 | -153.471* | 3.799 | -0.409 | 0.275** | 16.952 | 0.960* | 0.222* |
| | (-0.58) | (-1.77) | (1.19) | (-0.84) | (2.27) | (1.10) | (1.75) | (1.86) |
| GS | | | 0.018*** | -0.000 | | | 0.014*** | -0.000 |
| | | | (12.49) | (-0.65) | | | (42.42) | (-1.08) |
| CRD | | | | 0.027*** | | | | 0.040*** |
| | | | | (3.80) | | | | (18.32) |
| Time effect | Yes | Yes | Yes | Yes | Yes | Yes | Yes | Yes |
| Industry effects | Yes | Yes | Yes | Yes | Yes | Yes | Yes | Yes |
| Regional effects | Yes | Yes | Yes | Yes | Yes | Yes | Yes | Yes |
| _cons | -4.992*** | -248.639 | -31.949*** | -3.654*** | -6.593*** | -620.052*** | -31.294*** | -4.916*** |
| | (-8.56) | (-0.56) | (-7.46) | (-5.29) | (-47.43) | (-11.62) | (-44.96) | (-29.62) |
| N | 473 | 473 | 473 | 473 | 10096 | 10096 | 10096 | 10096 |
| $r^2$ | 0.704 | 0.037 | 0.622 | 0.715 | 0.654 | 0.044 | 0.481 | 0.667 |

If it falls within one of the six sectors of focus, then the listed company is defined as a more polluting company and the rest are defined as less polluting. In this paper, pollute is defined as a dummy variable, and the value of pollute is recorded as 1 if it is a highly polluting enterprise and 0 if it is a low-polluting enterprise.

Table 10 presents the results of grouped regressions for highly and lightly polluting firms. The study incorporates investor sentiment as a mediating variable, with models (1)-(4) focusing on a sample of significantly polluting enterprises, whereas models (5)-(8) analyze a sample of somewhat polluting firms. Among them, The mediating effect of investor sentiment is stronger for enterprises with low pollution level, while the air quality of heavily polluting companies is not significant for investor sentiment and there is no mediating effect.

The regression findings for heavy and light polluters, with government attention and R&D spending as mediating factors, are presented in Table 11. Among these businesses, the mediating impact is more prominent in the enterprises that exhibit higher levels of pollution. Conversely, in the firms with lower levels of pollution, no mediating effect is observed. When comparing model (2) with model (6), it is seen that government concerns exhibit a higher level of sensitivity to variations in air quality for severely polluting enterprises as opposed to weakly polluting firms.

**Table 10. Heterogeneity analysis of polluting firms in terms of investor sentiment.**

| Variables | (1) | (2) | (3) | (4) | (5) | (6) | (7) | (8) |
|---|---|---|---|---|---|---|---|---|
| | Pollute = 1 | | | | Pollute = 0 | | | |
| | TFP | Sent | CRD | TFP | TFP | Sent | CRD | TFP |
| AQI | -0.078*** | 0.025 | -0.218** | -0.071*** | -0.039*** | 0.075*** | -0.023 | -0.038*** |
| | (-3.45) | (0.70) | (-2.00) | (-3.17) | (-3.76) | (5.81) | (-0.44) | (-3.74) |
| ALE | 0.046 | 0.091 | -1.382*** | 0.100 | 0.300*** | 0.027 | -1.018*** | 0.339*** |
| | (0.55) | (0.94) | (-3.48) | (1.22) | (8.95) | (0.80) | (-5.95) | (10.31) |
| Size | 1.531*** | -0.189*** | 4.189*** | 1.373*** | 1.504*** | -0.122*** | 4.927*** | 1.314*** |
| | (43.77) | (-2.67) | (24.94) | (33.51) | (104.71) | (-5.16) | (67.21) | (76.14) |
| Cash | 0.222 | 0.099 | 0.580 | 0.200 | -0.052 | 0.210*** | 0.695** | -0.080 |
| | (1.39) | (0.80) | (0.75) | (1.27) | (-0.91) | (4.67) | (2.37) | (-1.41) |
| FCF | 1.018*** | 0.165 | 2.449*** | 0.944*** | 1.176*** | 0.237*** | 3.280*** | 1.042*** |
| | (5.66) | (1.49) | (2.82) | (5.30) | (16.65) | (5.67) | (9.06) | (14.92) |
| TCD | 0.008 | 0.015*** | -0.051 | 0.010 | 0.000 | 0.004*** | -0.068*** | 0.003 |
| | (0.88) | (3.36) | (-1.22) | (1.11) | (0.09) | (3.18) | (-4.55) | (1.07) |
| CYP | 0.004 | 0.022 | 0.035 | 0.003 | 0.007*** | 0.013** | 0.070*** | 0.004** |
| | (0.96) | (1.48) | (1.58) | (0.70) | (3.71) | (2.49) | (7.43) | (2.31) |
| GDP | 0.166 | -0.143 | 0.181 | 0.152 | 0.304** | -0.012 | 2.016*** | 0.228* |
| | (0.61) | (-0.94) | (0.14) | (0.57) | (2.33) | (-0.17) | (3.03) | (1.78) |
| Sent | | | 0.014 | -0.071 | | | 0.565*** | 0.010 |
| | | | (0.05) | (-1.12) | | | (4.12) | (0.39) |
| CRD | | | | 0.038*** | | | | 0.039*** |
| | | | | (7.10) | | | | (19.13) |
| Time effect | Yes | Yes | Yes | Yes | Yes | Yes | Yes | Yes |
| Industry effects | Yes | Yes | Yes | Yes | Yes | Yes | Yes | Yes |
| Regional effects | Yes | Yes | Yes | Yes | Yes | Yes | Yes | Yes |
| _cons | -7.165*** | 1.554** | -39.266*** | -5.686*** | -6.535*** | 0.774*** | -47.086*** | -4.715*** |
| | (-17.85) | (2.18) | (-20.38) | (-12.73) | (-45.00) | (3.24) | (-63.54) | (-27.53) |
| N | 1510 | 1510 | 1510 | 1510 | 9059 | 9059 | 9059 | 9059 |
| $r^2$ | 0.661 | 0.025 | 0.390 | 0.673 | 0.655 | 0.026 | 0.396 | 0.669 |

**Air quality, key regulated enterprises and total factor productivity of enterprises.** Based on the degree of regulation a listed company is subject to, this paper sets whether it is a key regulated company (Control) as one of the dummy variables. Set the value of Control to 1 for key regulated companies; set the value of Control to 0 for non-key regulated companies. Table 12 shows the regression results of whether the enterprises grouped by investor sentiment as the mediating variable are under centralized supervision or not, where models (1)-(4) are the samples of enterprises under centralized supervision, and models (5)-(8) are the samples of enterprises not under centralized supervision. There is a significant mediating effect for non-focused regulated firms, while there is no mediating effect for focused regulated firms. Table 13 shows the results of the regressions for the grouping of firms that are or are not key regulators with the government's interest as the mediating variable and the model set as mentioned above. There is no significant mediating effect of government attention for priority regulated firms, while there is a significant mediating effect of government attention for non-priority firms.

## Discussion

This paper investigates the impact of air pollution on the total factor productivity of listed companies, which is a multifaceted issue. This paper uses more representative data from listed

**Table 11. Heterogeneity analysis of polluting enterprises of concern to the government.**

| Variables | (1) | (2) | (3) | (4) | (5) | (6) | (7) | (8) |
|---|---|---|---|---|---|---|---|---|
| | Pollute = 1 | | | | Pollute = 0 | | | |
| | TFP | GS | CRD | TFP | TFP | GS | CRD | TFP |
| AQI | -0.078*** | 23.618** | -0.222** | -0.070*** | -0.039*** | 3.901 | 0.006 | -0.038*** |
| | (-3.45) | (2.32) | (-2.06) | (-3.15) | (-3.76) | (1.16) | (0.14) | (-3.76) |
| ALE | 0.046 | -41.659* | -1.327*** | 0.099 | 0.300*** | -5.379 | -0.611*** | 0.338*** |
| | (0.55) | (-1.76) | (-3.37) | (1.22) | (8.95) | (-0.72) | (-3.99) | (10.25) |
| Size | 1.531*** | 111.271*** | 3.820*** | 1.358*** | 1.504*** | 59.666*** | 3.125*** | 1.321*** |
| | (43.77) | (5.66) | (21.11) | (31.86) | (104.71) | (10.39) | (41.38) | (74.56) |
| Cash | 0.222 | -18.672 | 0.737 | 0.209 | -0.052 | -15.846 | 0.556** | -0.080 |
| | (1.39) | (-0.62) | (0.97) | (1.33) | (-0.91) | (-1.61) | (2.12) | (-1.41) |
| FCF | 1.018*** | 61.850** | 2.010** | 0.905*** | 1.176*** | 30.145*** | 2.421*** | 1.047*** |
| | (5.66) | (2.23) | (2.34) | (5.08) | (16.65) | (3.26) | (7.50) | (15.04) |
| TCD | 0.008 | 1.662 | -0.044 | 0.010 | 0.000 | 1.028*** | -0.060*** | 0.003 |
| | (0.88) | (1.40) | (-1.07) | (1.15) | (0.09) | (2.93) | (-4.47) | (1.06) |
| CYP | 0.004 | -0.289 | 0.025 | 0.003 | 0.007*** | 2.016* | 0.046*** | 0.004** |
| | (0.96) | (-0.08) | (1.16) | (0.59) | (3.71) | (1.76) | (5.46) | (2.34) |
| GDP | 0.166 | -12.529 | -0.003 | 0.150 | 0.304** | 11.830 | 1.513** | 0.228* |
| | (0.61) | (-0.32) | (-0.00) | (0.56) | (2.33) | (0.71) | (2.54) | (1.78) |
| GS | | | 0.005*** | 0.000 | | | 0.016*** | -0.000* |
| | | | (5.20) | (1.28) | | | (47.96) | (-1.74) |
| CRD | | | | 0.037*** | | | | 0.040*** |
| | | | | (6.87) | | | | (17.89) |
| Time effect | Yes | Yes | Yes | Yes | Yes | Yes | Yes | Yes |
| Industry effects | Yes | Yes | Yes | Yes | Yes | Yes | Yes | Yes |
| Regional effects | Yes | Yes | Yes | Yes | Yes | Yes | Yes | Yes |
| _cons | -7.165*** | -1.1e+03*** | -35.868*** | -5.547*** | -6.535*** | -547.337*** | -30.561*** | -4.774*** |
| | (-17.85) | (-5.67) | (-17.77) | (-12.07) | (-45.00) | (-9.90) | (-40.96) | (-27.33) |
| N | 1510 | 1510 | 1510 | 1510 | 9059 | 9059 | 9059 | 9059 |
| $r^2$ | 0.661 | 0.060 | 0.401 | 0.673 | 0.655 | 0.042 | 0.518 | 0.669 |

companies for verification and finds that air pollution significantly suppresses total factor productivity of enterprises, with an impact coefficient of -0.038. Compared with existing research [7], the conclusion is consistent. However, when PM2.5 is used to represent air pollution, the impact coefficient will increase, but PM2.5 is only one item in the Air Quality Index (AQI). Therefore, the AQI method of assessing air pollution yields more realistic findings. Scholars have expressed air quality improvement through AQI data [40], and obtained results from the Chinese industrial enterprise database. The impact of air quality improvement on total factor productivity of enterprises is significantly positive, which confirms the accuracy of the conclusions of this paper from another perspective.

The impact of air pollution on investor sentiment has been confirmed [21]. The deterioration of air pollution can bring pessimistic emotions to investors, which can reduce corporate investment [20]. However, due to government regulation and corporate environmental responsibility, air pollution may force companies to increase investment in green technology innovation [41]. The empirical results of this paper show that air pollution significantly suppresses investor sentiment, but investor sentiment significantly increases corporate R&D investment, and the increase in R&D investment further improves the total factor productivity of enterprises. Therefore, strengthening government supervision and focusing on improving

**Table 12. Heterogeneity analysis of whether investor sentiment is a key regulatory firm.**

| Variables | (1) | (2) | (3) | (4) | (5) | (6) | (7) | (8) |
|---|---|---|---|---|---|---|---|---|
| | Control = 1 | | | | Control = 0 | | | |
| | TFP | Sent | CRD | TFP | TFP | Sent | CRD | TFP |
| AQI | -0.005 | 0.100*** | -0.208* | 0.002 | -0.047*** | 0.060*** | -0.016 | -0.046*** |
| | (-0.29) | (2.71) | (-1.76) | (0.11) | (-4.32) | (4.20) | (-0.32) | (-4.34) |
| ALE | 0.209*** | 0.152 | -2.017*** | 0.275*** | 0.283*** | -0.002 | -1.030*** | 0.327*** |
| | (3.37) | (1.57) | (-5.00) | (4.51) | (7.92) | (-0.04) | (-6.26) | (9.31) |
| Size | 1.461*** | 0.046 | 5.985*** | 1.264*** | 1.499*** | -0.162*** | 4.369*** | 1.314*** |
| | (55.66) | (0.55) | (34.98) | (40.02) | (94.80) | (-6.19) | (60.04) | (70.47) |
| Cash | 0.077 | 0.026 | -0.005 | 0.076 | -0.042 | 0.231*** | 0.799*** | -0.076 |
| | (0.65) | (0.20) | (-0.01) | (0.65) | (-0.68) | (4.76) | (2.84) | (-1.26) |
| FCF | 1.296*** | 0.157 | 1.295 | 1.246*** | 1.117*** | 0.248*** | 3.510*** | 0.965*** |
| | (10.44) | (1.63) | (1.59) | (10.20) | (14.41) | (5.23) | (9.82) | (12.58) |
| TCD | -0.004 | 0.024*** | -0.124*** | 0.000 | 0.002 | 0.003** | -0.050*** | 0.004 |
| | (-0.60) | (5.62) | (-3.06) | (0.07) | (0.56) | (2.26) | (-3.50) | (1.30) |
| CYP | -0.000 | 0.038** | 0.059*** | -0.002 | 0.008*** | 0.011* | 0.067*** | 0.005*** |
| | (-0.12) | (2.51) | (2.62) | (-0.70) | (4.11) | (1.73) | (7.37) | (2.72) |
| GDP | 0.143 | -0.051 | 0.847 | 0.119 | 0.369** | -0.016 | 1.959*** | 0.286** |
| | (0.74) | (-0.36) | (0.67) | (0.63) | (2.52) | (-0.19) | (2.91) | (1.99) |
| Sent | | | 0.345 | 0.016 | | | 0.454*** | -0.004 |
| | | | (1.04) | (0.32) | | | (3.46) | (-0.16) |
| CRD | | | | 0.033*** | | | | 0.042*** |
| | | | | (10.69) | | | | (17.94) |
| Time effect | Yes | Yes | Yes | Yes | Yes | Yes | Yes | Yes |
| Industry effects | Yes | Yes | Yes | Yes | Yes | Yes | Yes | Yes |
| Regional effects | Yes | Yes | Yes | Yes | Yes | Yes | Yes | Yes |
| _cons | -5.845*** | -1.063 | -57.214*** | -3.957*** | -6.574*** | 1.217*** | -41.575*** | -4.812*** |
| | (-21.90) | (-1.27) | (-32.88) | (-12.57) | (-40.63) | (4.63) | (-55.83) | (-25.76) |
| N | 2412 | 2412 | 2412 | 2412 | 8157 | 8157 | 8157 | 8157 |
| $r^2$ | 0.681 | 0.039 | 0.436 | 0.696 | 0.630 | 0.028 | 0.368 | 0.644 |

corporate environmental awareness are important measures to improve the total factor productivity of enterprises.

Air pollution will increase the debt burden of local governments [42], therefore, the role of government attention in air pollution and total factor productivity of enterprises is a key issue. In the results of this paper, air pollution will attract significant attention from the government, leading to an increase in government subsidies for enterprises, thereby increasing R&D investment and improving total factor productivity of enterprises. To a certain extent, it can compensate for the crowding out effect of air pollution on enterprise research and development investment [43].

The nature of enterprises varies, and research results may also vary. The political landscape varies across enterprises with distinct property rights. Private firms are subject to a more stringent level of oversight in comparison to state-owned enterprises. The government will exert more pressure for more environmental responsibility to be taken up by private companies. In the context of increased atmospheric pollution, investor sentiment towards non-state enterprises is significantly higher, which is conducive to an increase in the stimulation of subsequent R&D investment by firms' innovative production, and also increases the level of subsequent R&D and innovation investment by firms, which in turn increases total factor

**Table 13. Heterogeneity analysis of whether the government is concerned about the key regulated enterprises.**

| Variables | (1) | (2) | (3) | (4) | (5) | (6) | (7) | (8) |
|---|---|---|---|---|---|---|---|---|
| | Control = 1 | | | | Control = 0 | | | |
| | TFP | GS | CRD | TFP | TFP | GS | CRD | TFP |
| AQI | -0.005 | 6.211 | -0.166 | 0.001 | -0.047*** | 5.696* | 0.007 | -0.046*** |
| | (-0.29) | (0.49) | (-1.51) | (0.07) | (-4.32) | (1.77) | (0.16) | (-4.34) |
| ALE | 0.209*** | -24.132 | -1.535*** | 0.271*** | 0.283*** | 1.816 | -0.687*** | 0.327*** |
| | (3.37) | (-0.85) | (-4.07) | (4.45) | (7.92) | (0.25) | (-4.60) | (9.30) |
| Size | 1.461*** | 79.567*** | 4.478*** | 1.280*** | 1.499*** | 59.348*** | 2.798*** | 1.314*** |
| | (55.66) | (3.01) | (25.15) | (39.73) | (94.80) | (10.77) | (37.03) | (68.34) |
| Cash | 0.077 | 1.237 | 0.622 | 0.064 | -0.042 | -16.003* | 0.550** | -0.076 |
| | (0.65) | (0.03) | (0.86) | (0.55) | (-0.68) | (-1.73) | (2.16) | (-1.26) |
| FCF | 1.296*** | 91.436*** | 0.655 | 1.262*** | 1.117*** | 29.836*** | 2.587*** | 0.964*** |
| | (10.44) | (3.22) | (0.87) | (10.40) | (14.41) | (3.30) | (7.99) | (12.60) |
| TCD | -0.004 | 3.837*** | -0.104*** | 0.000 | 0.002 | 0.674** | -0.043*** | 0.004 |
| | (-0.60) | (2.70) | (-2.76) | (0.05) | (0.56) | (2.09) | (-3.34) | (1.31) |
| CYP | -0.000 | 1.356 | 0.049** | -0.002 | 0.008*** | 2.399** | 0.040*** | 0.005*** |
| | (-0.12) | (0.30) | (2.32) | (-0.68) | (4.11) | (2.03) | (4.86) | (2.73) |
| GDP | 0.143 | 40.952 | -0.091 | 0.134 | 0.369** | 22.316 | 1.687*** | 0.287** |
| | (0.74) | (0.98) | (-0.08) | (0.71) | (2.52) | (1.34) | (2.77) | (2.00) |
| GS | | | 0.013*** | -0.000** | | | 0.015*** | -0.000 |
| | | | (19.04) | (-2.41) | | | (42.37) | (-0.05) |
| CRD | | | | 0.036*** | | | | 0.042*** |
| | | | | (10.86) | | | | (16.27) |
| Time effect | Yes | Yes | Yes | Yes | Yes | Yes | Yes | Yes |
| Industry effects | Yes | Yes | Yes | Yes | Yes | Yes | Yes | Yes |
| Regional effects | Yes | Yes | Yes | Yes | Yes | Yes | Yes | Yes |
| _cons | -5.845*** | -730.123*** | -43.373*** | -4.099*** | -6.574*** | -557.784*** | -27.200*** | -4.815*** |
| | (-21.90) | (-2.85) | (-24.40) | (-12.82) | (-40.63) | (-10.55) | (-36.09) | (-25.19) |
| N | 2412 | 2412 | 2412 | 2412 | 8157 | 8157 | 8157 | 8157 |
| r² | 0.681 | 0.039 | 0.436 | 0.696 | 0.630 | 0.028 | 0.368 | 0.644 |

productivity. In order to protect the favorable position of state-owned enterprises in market competition and the realization of benefits, local governments can intervene through direct monopoly, entrusting enterprises and other methods. Therefore, government concern does not in itself act as a pathway of influence between air pollution and the total factor productivity.

## Conclusions and policy recommendations

### Conclusion

This study utilizes authentic and reliable data from Shanghai and Shenzhen to conduct a research analysis on the influence of air pollution on the total factor productivity. The results are as follows:

1. The quality of air has a substantial negative impact on the total factor productivity. Nevertheless, it is worth noting that air pollution exerts a substantial positive impact on investor mood, so stimulating enterprises to expand their research and development spending as a means to enhance their overall factor productivity. The issue of air pollution has the

potential to garner significant government focus, resulting in heightened research and development investments.

2. Furthermore, this study also examines the variety of the enterprise's nature, its potential contribution to pollution, and its role as a focal point for government oversight. As a result, the following findings are drawn:

The findings indicate that air quality plays a mediating role in the relationship between investor sentiment and R&D investment on firms' total factor productivity, with the exception of state-owned firms. Additionally, the mediating effects of government concern and R&D investment on the relationship between air quality and firms' total factor productivity are observed exclusively in non-state-owned firms.

The impact of air quality on investor sentiment is significant, this mediating effect is more pronounced for light-polluting firms compared to heavy polluting enterprises. The analysis reveals a noteworthy mediating effect of government concern as a variable in the case of highly polluting companies, while no mediating effect is observed in the case of lightly polluting firms. Furthermore, it is observed that government concern is more responsive to variations in air quality for heavily polluting firms as compared to lightly polluting firms.

Investor sentiment plays a substantial role as a mediating variable for non-focused regulated enterprises, but not for focused regulated firms. The mediating impact of government attention is not shown to be substantial for businesses subject to focal regulation, however it is found to be significant for enterprises not subject to focal regulation.

## Policy recommendations

**1. Optimize and control environmental regulations.**    The optimization and control of environmental laws are crucial for the improvement of total factor productivity and sustainable development. The rationalization of environmental regulatory policies can also be achieved through the optimization and upgrading of industrial structure, using environmental regulation as a mediator to promote the improvement of total factor productivity. In addition, utilizing modern technologies such as artificial intelligence to improve environmental governance efficiency and improve tax collection and management related to environmental protection. It is crucial to consider the heterogeneity of the impact of different types of enterprises and regions. For example, state-owned and large enterprises are better able to adapt to environmental regulations, and the impact of these regulations on TFP may vary by regiond industry.

**2. Strengthen enterprise digital transformation.**    Digitization greatly affects investor sentiment by expanding the acquisition of market data and promoting real-time communication. Through digital transformation, it actively shapes investor perception and market trends. Especially for high polluting enterprises, deepening the integration of digital technology to enhance green technology innovation and corporate social responsibility can improve TFP both internally externally.

**3. Strengthen the intensity of government subsidies.**    Government concern refers to the involvement of government-subsidized enterprises in facilitating a connection between air quality and total factor production. The reflection may be observed through the examination of the subsequent two locations. (1) It is recommended that the government enhance enterprise oversight, as the data collected during supervision directly impacts the subsidies granted to these enterprises. This approach can serve as an incentive mechanism to foster innovation within the business sector. (2) Enterprises should capitalize on the momentum generated by the ongoing IT revolution. By closely monitoring market demands, they can strategically focus

on upgrading and innovating key technologies. This endeavor will ultimately enhance their competitiveness in the global market.

## Limitation, and future work

However, this paper has some limitations in data selection and classification. Due to the fact that the research data in this paper is a combination of enterprise level and city level data, there has been a serious lack of enterprise data since 2019, which has affected empirical analysis.

## Supporting information

**S1 Table. The data of all variables.**
(XLSX)

## Acknowledgments

Thank you to all the reviewers for their suggested revisions, which are of great significance to this paper.

## Author Contributions

**Conceptualization:** Jialiang Yang, Wen Yin.

**Data curation:** Jialiang Yang.

**Formal analysis:** Jialiang Yang.

**Funding acquisition:** Jialiang Yang.

**Investigation:** Jialiang Yang.

**Methodology:** Jialiang Yang.

**Project administration:** Jialiang Yang.

**Resources:** Jialiang Yang, Wen Yin.

**Software:** Jialiang Yang.

**Supervision:** Wen Yin.

**Writing – original draft:** Jialiang Yang.

**Writing – review & editing:** Jialiang Yang, Wen Yin.

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
