## [Decision Letter · Decision Letter 0]

8 Mar 2024

PONE-D-24-04688How air pollution affects corporate total factor productivity?PLOS ONE

Dear Dr. Yin,

Thank you for submitting your manuscript to PLOS ONE. After careful consideration, we feel that it has merit but does not fully meet PLOS ONE’s publication criteria as it currently stands. Therefore, we invite you to submit a revised version of the manuscript that addresses the points raised during the review process.

We look forward to receiving your revised manuscript.

Kind regards,

Chaohai Shen

Academic Editor

PLOS ONE

Journal Requirements:

"This research was funded by [Top-notch Academic Programs Project of Jiangsu Higher Education Institutions (Provincial First-class Specialty Construction of Energy Economy in Jingjiang College of Jiangsu University)]."

"This research was funded by [Top-notch Academic Programs Project of Jiangsu Higher Education Institutions (Provincial First-class Specialty Construction of Energy Economy in Jingjiang College of Jiangsu University)]."

"This research was funded by [Top-notch Academic Programs Project of Jiangsu Higher Education Institutions (Provincial First-class Specialty Construction of Energy Economy in Jingjiang College of Jiangsu University)]."

5. We note that [Figure 1] in your submission contain [map/satellite] images which may be copyrighted. All PLOS content is published under the Creative Commons Attribution License (CC BY 4.0), which means that the manuscript, images, and Supporting Information files will be freely available online, and any third party is permitted to access, download, copy, distribute, and use these materials in any way, even commercially, with proper attribution. For these reasons, we cannot publish previously copyrighted maps or satellite images created using proprietary data, such as Google software (Google Maps, Street View, and Earth). For more information, see our copyright guidelines: http://journals.plos.org/plosone/s/licenses-and-copyright.

Additional Editor Comments:

Dear Authors,

I have received the required number of reviewers' reports. Based on their comments and my own justification after reading the paper, I think you have to make substantial revisions.

In particular, we consider the literature part is very weak with the hypotheses not well developed. You should completely rewrite the literature part with most recent and classical papers properly cited. These papers were usually published in decently good international peer-reviewed journals.

Also, the empirical part should be further improved by clearly justifying the validity of the model.

There are more issues raised in the reviewers' reports. Please carefully read them and make revisions accordingly.

Sincerely,

Reviewers' comments:

Reviewer's Responses to Questions

**Comments to the Author**

1. Is the manuscript technically sound, and do the data support the conclusions?

Reviewer #1: No

Reviewer #2: Partly

2. Has the statistical analysis been performed appropriately and rigorously? 

Reviewer #1: Yes

Reviewer #2: No

3. Have the authors made all data underlying the findings in their manuscript fully available?

Reviewer #1: No

Reviewer #2: No

4. Is the manuscript presented in an intelligible fashion and written in standard English?

Reviewer #1: No

Reviewer #2: Yes

5. Review Comments to the Author

Reviewer #1: The article has a poor analysis of the literature, and there is no research gap. The hypotheses were formulated based on what? Why was this methodology adopted? what does this mean? The lack of these assumptions precludes further analysis

Reviewer #2: This study investigates the impact mechanism of air quality on corporate total factor productivity by selecting listed firms in China. Although it is very interesting, there are many problems. Significant revisions to this study are required before it can be considered for publication.

1. Abstracts are excessively long. It fails to succinctly elucidate the purpose or gaps in the research.

2. The Introduction section is poorly designed. The gaps between this study and existing studies are not visible in the Introduction. Meanwhile, this section fails to give the research innovation.

3. The Literature Review is weak in supporting the research hypotheses. Firstly, the Literature Review of this study fails to support the research hypotheses well and needs to be further strengthened. Secondly, the formulation of research hypotheses needs not only the support of authoritative literature but also the support of classical theories.

4. Does the interactive effect of research hypothesis 2 and research hypothesis 3 exist? For example, the government's attention impacts investor sentiment. Therefore, this section needs to be redesigned.

5. The "Ethics Statement" on page 7 does not seem appropriate here.

6. Research data is too old. The author mentions in the limitations section, “Due to the availability of data, the data in this paper is only up to 2019.” Is this an insurmountable challenge? If possible, please update the research data and reiterate the limitations.

7. This study fails to provide descriptive statistics for all variables, which weakens the scientific validity of the empirical research.

8. The Discussion section is extremely poor. The present Discussion simply analyses the empirical results and does not compare them with existing studies to clarify how this study contributes to, adds to, or differs from existing studies. It is more a test of heterogeneity than a discussion.

9. Policy recommendations need to be more specific, nuanced, and feasible. In addition, limitations and future research need to be more detailed.

10. English level needs to be improved. It has grammatical errors and presentation errors. In addition, this study needs to pay attention to the use of long sentences, acronyms, and proper nouns.

6. PLOS authors have the option to publish the peer review history of their article (what does this mean?). If published, this will include your full peer review and any attached files.

Reviewer #1: No

Reviewer #2: No

---

## [Author Response · Author response to Decision Letter 0]

20 Apr 2024

Dear Professor Shen

Thank you very much to the editor and reviewers for their valuable revision suggestions, which are of great significance to our paper. According to the questions from the teachers, the response is as follows:

Question: Please include this amended Role of Funder statement in your cover letter; we will change the online submission form on your behalf ?

Answer: The funders had no role in study design, data collection and analysis, decision to publish, or preparation of the manuscript. We have deleted relevant information in the Acknowledgement.

Question: Funding information should not appear in the Acknowledgments section or other areas of your manuscript. We will only publish funding information present in the Funding Statement section of the online submission form. Please remove any funding-related text from the manuscript and let us know how you would like to update your Funding Statement.

Answer: We have removed the funding-related text in the Acknowledgement section. Our amended statements as follows: This research was funded by [Top-notch Academic Programs Project of Jiangsu Higher Education Institutions (Provincial First-class Specialty Construction of Energy Economy in Jingjiang College of Jiangsu University)].

Question: Please ensure that you have an ORCID iD and that it is validated in Editorial Manager.

Answer: The corresponding author already has it (ID: 0009-0002-4234-8737).

Question: We note that [Figure 1] in your submission contain [map/satellite] images which may be copyrighted.

Answer: Thank you for your correction, We have removed the Figure 1.

Question: Have the authors made all data underlying the findings in their manuscript fully available?

Answer: We have supplemented the provided data, please refer to Table S1 for details.

Here are the responses to the two reviewer teachers.

Reviewer #1:

Question: The article has a poor analysis of the literature, and there is no research gap. The hypotheses were formulated based on what? Why was this methodology adopted? what does this mean? The lack of these assumptions precludes further analysis.

Answer: Thank you for your correction, we have rewritten the introduction and literature review sections, summarized the latest research progress, proposed innovations in this article, and proposed hypotheses based on existing research conclusions and gaps.

Reviewer #2: 

Question1: Abstracts are excessively long. It fails to succinctly elucidate the purpose or gaps in the research.

Answer: Thank you for your correction. We have rewritten the abstract section, shortened the length, and clarified the purpose.

Question 2: The Introduction section is poorly designed. The gaps between this study and existing studies are not visible in the Introduction. Meanwhile, this section fails to give the research innovation.

Answer: Thank you for your correction. We have completely rewritten the introduction section, citing current research conclusions and pointing out the innovative points of this article.

Question 3: The Literature Review is weak in supporting the research hypotheses. Firstly, the Literature Review of this study fails to support the research hypotheses well and needs to be further strengthened. Secondly, the formulation of research hypotheses needs not only the support of authoritative literature but also the support of classical theories.

Answer: In response to your question, we have made modifications by summarizing the latest existing literature as the basis for the research design of this article, and proposing a new research path for this article.

Question 4: Does the interactive effect of research hypothesis 2 and research hypothesis 3 exist? For example, the government's attention impacts investor sentiment. Therefore, this section needs to be redesigned.

Answer: Thank you for your correction. We have redesigned this part of the hypothesis and analyzed the two paths separately to avoid any interaction. At the same time, this article has conducted endogeneity and robustness tests on the empirical results to ensure their rationality.

Question 5: The "Ethics Statement" on page 7 does not seem appropriate here.

Answer: Thank you for your correction. We fully agree with your point of view and have deleted this section.

Question 6: Research data is too old. The author mentions in the limitations section, “Due to the availability of data, the data in this paper is only up to 2019.” Is this an insurmountable challenge? If possible, please update the research data and reiterate the limitations.

Answer: Thank you for your correction. Due to a significant lack of data on listed companies in later years, in order to ensure a sufficient sample size, we have chosen the current year for analysis. Based on your suggestion, we have made modifications to the limitations. Thank you again.

Question 7: This study fails to provide descriptive statistics for all variables, which weakens the scientific validity of the empirical research.

Answer: Thank you for your correction. We fully agree and have added a descriptive statistics section, as shown in Table 2.

Question 8: The Discussion section is extremely poor. The present Discussion simply analyses the empirical results and does not compare them with existing studies to clarify how this study contributes to, adds to, or differs from existing studies. It is more a test of heterogeneity than a discussion.

Answer: Thank you for your correction. We have added a new discussion section and changed the previous discussion section to heterogeneity analysis.

Question 9: Policy recommendations need to be more specific, nuanced, and feasible. In addition, limitations and future research need to be more detailed.

Answer: Thank you for your correction. We have rewritten the policy recommendations section to make it more targeted. Meanwhile, the limitations have also been rewritten.

Question 10: English level needs to be improved. It has grammatical errors and presentation errors. In addition, this study needs to pay attention to the use of long sentences, acronyms, and proper nouns.

Answer: Thank you for your correction. We have improved our English.

Best wishes

---

## [Decision Letter · Decision Letter 1]

7 May 2024

How air pollution affects corporate total factor productivity?

PONE-D-24-04688R1

Dear Dr. Yin,

We’re pleased to inform you that your manuscript has been judged scientifically suitable for publication and will be formally accepted for publication once it meets all outstanding technical requirements.

Kind regards,

Chaohai Shen

Academic Editor

PLOS ONE

Additional Editor Comments (optional):

Dear Authors,

I have received the required number of reviewer reports. Based on their comments and my own justification, I think the revised version can be accepted for publication.

Reviewers' comments:

Reviewer's Responses to Questions

**Comments to the Author**

1. If the authors have adequately addressed your comments raised in a previous round of review and you feel that this manuscript is now acceptable for publication, you may indicate that here to bypass the “Comments to the Author” section, enter your conflict of interest statement in the “Confidential to Editor” section, and submit your "Accept" recommendation.

Reviewer #1: All comments have been addressed

Reviewer #2: All comments have been addressed

2. Is the manuscript technically sound, and do the data support the conclusions?

Reviewer #1: Yes

Reviewer #2: Yes

3. Has the statistical analysis been performed appropriately and rigorously? 

Reviewer #1: Yes

Reviewer #2: Yes

4. Have the authors made all data underlying the findings in their manuscript fully available?

Reviewer #1: Yes

Reviewer #2: No

5. Is the manuscript presented in an intelligible fashion and written in standard English?

Reviewer #1: Yes

Reviewer #2: Yes

6. Review Comments to the Author

Reviewer #1: corrections were made in accordance with the comments, I have no other comments. the article is suitable for publication

Reviewer #2: (No Response)

7. PLOS authors have the option to publish the peer review history of their article (what does this mean?). If published, this will include your full peer review and any attached files.

Reviewer #1: No

Reviewer #2: No

---

## [Editor Report · Acceptance letter]

15 May 2024

PONE-D-24-04688R1 

PLOS ONE

Dear Dr. Yin, 

I'm pleased to inform you that your manuscript has been deemed suitable for publication in PLOS ONE. Congratulations! Your manuscript is now being handed over to our production team.

Kind regards, 

on behalf of

Dr. Chaohai Shen 

Academic Editor

PLOS ONE